# Pollution Assessment and Source Apportionment of Soil Heavy Metals in a Coastal Industrial City, Zhejiang, Southeastern China

**DOI:** 10.3390/ijerph19063335

**Published:** 2022-03-11

**Authors:** Shiyi Wang, Yanbin Zhang, Jieliang Cheng, Yi Li, Feng Li, Yan Li, Zhou Shi

**Affiliations:** 1School of Public Affairs, Institute of Land Science and Property, Zhejiang University, Hangzhou 310058, China; shiyiwang@zju.edu.cn (S.W.); labyr1nth@zju.edu.cn (Y.L.); 2Zhejiang Land Consolidation and Rehabilitation Center, Hangzhou 310007, China; zhangyanbin919@163.com; 3Zhejiang Cultivated Land Quality and Fertilizer Management Station, Hangzhou 310020, China; chengjl2000@163.com; 4College of Materials and Environmental Engineering, Hangzhou Dianzi University, Hangzhou 310058, China; lifeng@hdu.edu.cn; 5Key Laboratory of Environment Remediation and Ecological Health, Ministry of Education, College of Environmental and Resource Sciences, Zhejiang University, Hangzhou 310058, China; shizhou@zju.edu.cn

**Keywords:** soil heavy metals, pollution assessment, source apportionment, industrial city

## Abstract

In this research, Ningbo City, a typical industrial city in southeastern China, was selected as the study area, and the concentrations of 12 heavy metals (Cd, Cr, Ni, Pb, Zn, Cu, Hg, As, Co, V, Se, and Mn) were measured at 248 sampling points. Pollution index methods were used to assess the status of soil heavy metal contamination, and the Positive Matrix Factorization (PMF) model and Unmix model were integrated to identify and apportion the sources of heavy metal contamination. The results indicated that nearly 70% of the study area was polluted by heavy metals, and that Ni, Cr, and Zn were the main enriched heavy metals. The five sources identified using the PMF model were a geological source, an atmospheric deposition source, a transportation emissions source, a mixed source of agriculture and industry, and a mixed source of geology and industry. The four sources identified using the Unmix model were a mixed source of geology, agriculture, and industry (14.27%); a transportation emissions source (4.76%); a geological source (14.7%); and a mixed source of geology and industry (66.28%). These results have practical significance, as they can help to carry out pollution source risk assessment and give priority to the management of pollution source control.

## 1. Introduction

Soil accommodates a large number of potentially toxic elements, including heavy metals [1]. Due to their accumulation, irreversibility, persistence, and high toxicity [2], excessive heavy metal concentrations lead to poor growth of plants and crops, a reduced number of animal communities [3], and threats to human health through food intake [4]. Thus, soil heavy metal pollution is directly related to economic development, social progress, and the well-being of residents [5]. In order to improve the safe utilization rate of contaminated land, it is necessary to alleviate further soil heavy metal pollution from the sources.

Previous studies have reported that soil heavy metal pollution is generally caused by natural and anthropogenic factors. The natural factors include the geological parent materials, which determine the background content of heavy metals [6,7]. Specifically, the accumulation potential of heavy metals is related to the soil type, texture, clay content, and chemical characteristics [8]. In addition, industrialization has exacerbated the impacts of anthropogenic factors, including untreated industrial emissions, irrational use of pesticides and fertilizers, atmospheric aerosol deposition, and vehicle exhausts [9,10,11]. Heavy metals related to the above processes enter the soil in the forms of effluent discharge, surface runoff, and air pollutants [12]. However, previous studies have failed to accurately and completely determine which pollution sources affect soil heavy metal enrichment [13,14,15]. Therefore, in this study, the potential sources of soil heavy metals are quantitatively and precisely identified.

Soil heavy metal source apportionment methods mainly include diffusion models and receptor models. Because diffusion models are limited by the difficulty of establishing a source emission inventory and the complexity of ecological and geochemical processes [16], receptor models have become a more common approach. Receptor models include the Chemical Mass Balance (CMB) model, Principal Component Analysis (PCA), the multi-linear engine, the Positive Matrix Factorization (PMF) model, and the Unmix model. Santos et al. used the CMB model to determine the contributions of six industrial sources [17], but the CMB model relies on the indigenous source component spectrum and cannot distinguish between different sources with similar chemical compositions. Wang et al. used PCA to identify the possible pollution sources of soil heavy metals in Jiangsu Province [18]. However, the component score obtained through PCA may be negative, and the complete contributions cannot be directly calculated. Yang et al. combined PCA with the multiple linear regression method to calculate the contributions of different pollution sources [19]. This method is more suitable for situations in which the concentrations of multiple heavy metals increase significantly both temporally and spatially. Chai et al. used the PMF model to analyze four pollution sources of eight heavy metals [20]. Liu et al. used the Unmix model to identify three pollution sources of five heavy metals in Beijing [21]. Considering the poor accuracy of using a single method due to the error generated by the algorithm or the influence of the sample size, the integration of the PMF model and the Unmix model has outstanding advantages in exploring the sources of heavy metal contamination in a study area [22,23].

Although industrialization is a major driving force of soil heavy metal pollution, the impact of industry has not been sufficiently revealed. Therefore, in this study, the types of heavy metal pollution exacerbated by specific industrial enterprises were investigated.

Ningbo is a chemical industry-dependent city in the southern part of the Yangtze River Delta in China. It contains eight state-level development zones, indicating a high level of industrialization. Industrial pollutants containing heavy metals spread to the surrounding soil by means of atmospheric deposition and eluviation [24]. Furthermore, the Zhenhai District has an irreplaceable industrial status in Ningbo, and the industrial structure, which is dominated by machinery, textile, and metal industries, has aggravated the soil heavy metal pollution. The objectives of this study were (1) to comprehensively and specifically reveal the pollution statuses of 12 soil heavy metals, including cadmium (Cd), chromium (Cr), nickel (Ni), lead (Pb), zinc (Zn), copper (Cu), mercury (Hg), arsenic (As), cobalt (Co), vanadium (V), selenium (Se), and manganese (Mn); (2) to identify the pollution sources and their contributions; and (3) to explore the influences of industrial enterprises on the heavy metal contents of the soil and propose targeted policy suggestions to prevent further soil pollution.

## 2. Materials and Methods

### 2.1. Study Area

To thoroughly dissect the impact of industry on soil heavy metal pollution, the Zhenhai District of Ningbo City, which is one of the most prosperous industrial regions in China [25], was selected as the study area. Ningbo (120°55′–122°16′ E, 28°51′–30°33′ N) is located in eastern China on the southeastern coast. It is one of the economic centers of Zhejiang Province, is the southern economic center in the Yangtze River Delta, and is a chemical industry base. Zhenhai has a land area of 246 km^2^ and a permanent population of 510,500. It had an urbanization rate of 65.8% at the end of 2020. Ranking 28th among the top 100 industrial regions in China, the Zhenhai District has a solid industrial foundation, and it is a major port for foreign trade [26]. In 2020, the total Gross Domestic Product (GDP) of Zhenhai was 103.01 billion yuan, and the ratio of the primary, secondary, and tertiary industries was 0.6:63.9:35.5 [26].

Construction land accounts for 40.54% of the study area. The population, traffic networks, industrial enterprises, and construction land in the study area are mainly distributed in the central and southern regions, which have gentle terrain. Human activities are frequent in these regions. In the northwest, flat areas are less abundant, the population and traffic network densities are relatively low; the soil is mainly red soil with a low pH value, and the main land-use types are woodland and arable land (Figure 1).

### 2.2. Data Collection and Preparation

In this study, samples were collected from 248 sampling points in 2016 using the stratified random sampling method, including 96 sampling sites in agricultural land, nine sites around large traffic arteries, and 143 sites in construction land. Among them, 111 sampling sites were located in major industrial (park) areas and their surrounding areas. A Global Positioning System (GPS) was used to determine the precise locations of the sampling sites. Surface soil samples were collected from the top 0–50 cm layer. Each soil sample was created by merging five subsamples gathered from five positions within a 5 m radius. In the sampling process, a shovel was used to cut an earthwork with a depth of 20 cm, which was larger than the amount of soil collected, and then wood or bamboo slices were used to collect the samples after removing the soil that came in contact with the shovel. Five hundred-gram samples, collected each time, were placed in a 500 mL brown sampling bottle and sealed with paraffin wax. The air-dried soil samples were sieved through a 2 mm nylon sieve in the laboratory. After preparation, the Cr, Ni, Pb, and Zn contents were measured via Inductively Coupled Plasma–Atomic Emission Spectrometry (ICP–AES), the Cd, Cu, Co, and V contents were determined via Inductively Coupled Plasma–Mass Spectrometry (ICP–MS), the Hg, As, and Se contents were determined via Microwave Digestion–Atomic Fluorescence Spectrometry (MD–AFS), and the Mn contents were determined via Atomic Absorption Spectrophotometry (AAS).

### 2.3. Statistical Analysis Method

Descriptive statistics, including the maximum, minimum, mean, standard deviation, median, skewness, kurtosis, and coefficient of variation, were calculated. The kurtosis is a measure of the flatness of data, and a high value indicates concentrated data [27]. The skewness is a measure of the symmetry of the data. A skewness tending to zero indicates that the data distribution is similar to a normal distribution [27]. The coefficient of variation is the ratio of the standard deviation to the mean, which can reflect the degree of variation of the sample data distribution. A coefficient of variation of less than 10% indicates weak variability, values of 10–100% indicate moderate variability, and values greater than 100% indicate strong variability. If the coefficient of variation is greater than 50%, there may be point source pollution caused by foreign substances [18].

The national risk screening value and background value are criteria for assessing the degree of pollution. The national risk screening value is set by the Ministry of Ecology and Environment of China (GB 15618-2018 and GB 36000-2018). Pollutants exceeding their corresponding values pose risks to crop growth and human health [20]. Due to the different geological characteristics and social environments of different regions, the background value can represent the heavy metal concentration that is not affected by human activities at a certain time and in a certain location, and it can be used to objectively evaluate the soil pollution status [28]. In this study, the background values for Ningshao Plain were used [29].

### 2.4. Pollution Index Method

Three pollution index methods, including the Single Pollution Index (SPI), Nemerow Integrated Pollution Index (NIPI), and Geo-accumulation Index (GI), were used in this study to assess the degree of pollution and the accumulation of heavy metals in the soil (Table 1). The SPI is used to evaluate the contamination degree of each particular heavy metal. The NIPI is calculated based on the SPI, and it can comprehensively reflect the average pollution level of various pollutants in the soil and highlight the damage caused by the most enriched pollutants [30]. The GI is an early developed and widely used method for studying heavy metal pollution in sediments and soils [31].

**Table 1 ijerph-19-03335-t001:** Classification standard for soil heavy metal pollution using three methods [32,33].

Single Pollution Index		Nemerow Integrated Pollution Index		Geo-Accumulation Index	
SPI ≤ 1	non-polluting	NIPI ≤ 0.7	background areas	GI < 0	no pollution
1 < SPI ≤ 2	mild pollution	0.7 < NIPI ≤ 1	warning areas	0 ≤ GI < 1	no pollution-moderate pollution
2 < SPI ≤ 3	moderate pollution	1 < NIPI ≤ 2	mildly polluted areas	1 ≤ GI < 2	moderate pollution
SPI > 3	severe pollution	2 < NIPI ≤ 3	moderately polluted areas	2 ≤ GI < 3	moderate pollution-heavy pollution
		NIPI > 3	heavily polluted areas	3 ≤ GI < 4	heavy pollution
				4 ≤ GI < 5	heavy pollution-extremely heavy pollution
				GI ≥ 5	extremely heavy pollution

### 2.5. Positive Matrix Factorization Model

The PMF model is an analytical tool used to quantify the sources of heavy metal pollution. Paatero and Tapper proposed the PMF non-negative factor model [34], which optimizes the error analysis of the data and ensures that no negative values appear in the results of factor matrix decomposition during the solving process so that the results have more practical physical significance [35]. Based on the least iterative square algorithm, the PMF model decomposes the original data matrix of the recipient (*X*) into a factor score matrix (*G*), a factor load matrix (*F*), and a residual matrix (*E*). The calculation formula of the PMF is as follows [36]:(1)$$X_{ij}=\overset{P}{\underset{k=1}{\sum }}\left(G_{ik}\times F_{kj}\right)\;+\;E_{ij}$$
where $X_{ij}$ is the content of element *j* in sample *i*, $G_{ik}$ is the relative contribution of the pollution source *k* in sample *i*, $F_{kj}$ is the characteristic value of pollution source *k* related to the concentration of element *j*, $E_{ij}$ is the residual, and *P* is the number of factors. All of the parameters are dimensionless.

The optimal matrices *G* and *F* decomposed using the PMF model should minimize the objective function *Q*, which is defined as follows [37]:(2)$$Q=\overset{n}{\underset{i=1}{\sum }}\overset{m}{\underset{j=1}{\sum }}{\left(\frac{E_{ij}}{U_{ij}}\right)}^{2}$$
where $U_{ij}$ is the uncertainty of the content of element *j* in sample *i*. The model can weight each individual data point and assign an appropriate uncertainty to each data point. When the concentration of the element is lower than or equal to the corresponding Method’s Detection Limit (*MDL*), the equation for calculating the uncertainty is as follows [38,39]:(3)$$U_{ij}=\frac{5}{6}\times MDL$$

### 2.6. Unmix Model

The Unmix model is a multi-receptor model developed by the US Environmental Protection Agency to solve the general mixing problem [40]. With the original dataset intact, the dataset itself can provide the required constraints and define unique solutions [41]. The operation of the Unmix model is simple, and there is no need to make complex adjustments to the parameters or to know the source component spectrum. In the Unmix model, the Singular Value Decomposition (SVD) method is used to reduce the dimension of the data space, and the objective is to determine the number of sources, the source compositions, and the contributions of the sources to each sample. Based on the above assumptions, the equation for calculating the content of element *j* is as follows [42]:(4)$$C_{ij}=\overset{m}{\underset{k=1}{\sum }}U_{jk}D_{ik}+S$$
where $C_{ij}$ is the content of element *j* in sample *I*, and $U_{jk}$ is the mass fraction of element *j* in source *k*, representing the composition of the source. $D_{ik}$ is the total amount of source *k* in sample *i*, denoting the contribution of the source. *S* is the standard deviation of each source’s composition.

Two metrics are employed in the Unmix model. MinRsq is yielded by the corresponding sources to denote the variance of each heavy metal, and MinSig/Noise is generated by the regression and error sum of the squares of the eigenvectors [43].

### 2.7. Data Analysis

The descriptive analysis and SPI and GI assessment were conducted using Microsoft Excel 2016 (Microsoft Corporation). The NIPI was calculated using Microsoft Excel 2016 (Microsoft Corporation), and the data selection, statistics, and spatial interpolation were then conducted using ArcGIS 10.4.1 (ESRI Inc., Redlands, CA, USA) and QGIS 3.6 (QGIS.ORG) in order to obtain the spatial distribution map. The PMF model was implemented using EPA PMF5.0. The Unmix model was implemented using Unmix 6.0.

## 3. Results

### 3.1. Descriptive Statistical Analysis

The descriptive statistics of the 12 heavy metal elements were obtained (Table 2). It was found that the mean contents of the 12 heavy metals were all greater than the corresponding background values, and the degrees to which they exceeded the background value were as follows: Hg > Ni > Cr > Zn > Cu > As > Cd > Co > Pb > V > Mn > Se. The maximum contents of the soil heavy metals in the study area exceeded the corresponding background values to varying degrees: Ni > Cr > Zn > Cd > Hg > Cu > Se > Pb > Co > Mn > As > V. Except for Pb and Co, the maximum values of the other heavy metals were greater than the national risk screening values (pH > 7.5). Analogously, the degrees to which they surpassed the standards were as follows: Zn > Cr > Ni > Cd > Hg > Cu > V > Se > Mn > As > Pb > Co. The mean Cr content far exceeded the national risk screening value, while the mean contents of the other heavy metals were within the national risk screening values, but they were all higher than the corresponding background values. It was preliminarily determined that the samples collected from the study area were polluted or even highly polluted, and Zn, Cr, and Ni were the heavy metals in the soil that posed the greatest risks.

Among the 12 soil heavy metals in the study area that were analyzed, the kurtosis and skewness values of Ni, Cr, and Zn were much higher than those of the other heavy metals, indicating that the distributions of the Ni, Cr, and Zn contents were concentrated at a level higher than the mean. In terms of the coefficients of variation, the variabilities of Ni, Cr, Zn, Cd, and Hg were strong, and the variabilities of the other elements were moderate. The average coefficient of variation of the heavy metal contents was 126.36%. The coefficients of variation of Pb, As, Co, V, and Mn were relatively low, indicating that they were less affected by anthropogenic activities.

### 3.2. Heavy Metal Pollution Assessment

Using the national risk screening value as the standard, the single pollution index in the study area was calculated (Figure 2). Among the 12 heavy metals, Cr exceeded the standard the most seriously, and 60.48% of the sampling sites were severely polluted by Cr. For Zn, 1.61% of the sampling sites were severely polluted, and 8.07% of the sampling sites were moderately polluted. For Cd, 0.40% of the sampling sites were severely polluted. Except for Cr, Zn, and Cd, none of the sampling sites were severely polluted. Furthermore, 38.71% of the sampling sites did not exhibit Cr contamination, and the proportion of sampling sites not contaminated by the other 11 heavy metals was greater than 90%. The As, Co, and V contents were within the national risk screening values.

Combined with the spatial interpolation method, the NIPI results calculated based on the SPI revealed that nearly 70% of the soil sites in the study area were contaminated by heavy metals. Moreover, except for small areas in the western, central, northern, southeastern, and southwestern regions, the rest of the soil sites were heavily polluted, accounting for 58.85% of the total study area and exhibiting a concentrated contiguous spatial distribution (Figure 3). By comprehensively considering the land-use types in the study area (Figure 1), it was determined that these regions also contained densely distributed industrial enterprises and traffic networks, with frequent anthropogenic activities. Due to its role of emphasizing the most prominent pollutants, the NIPI results indicated more severe pollution than the SPI, reflecting the imbalance between the pollution degrees of the heavy metals.

According to the GI (Figure 4), some of the sampling sites had extremely heavy Cr, Ni, and Zn pollution, each of which accounted for 0.40% of the total sampling sites. In contrast, the sampling sites with heavy Cd, Zn, Cu, and Hg pollution accounted for 1.21%, 0.81%, 0.40%, and 0.81% of the total sampling sites, respectively. In general, Cu had the largest proportion of contaminated sample sites, followed by Ni and Cr, all of which accounted for more than 60%. The degrees of V, Se, and Mn pollution were relatively small, 80% of the sampling points were within the pollution-free range, and there was no heavy pollution by these elements.

### 3.3. Source Apportionment Based on the PMF Model

The PMF model was used for the source apportionment of heavy metals. The number of factors was set to four to seven and the operation times was set as 20 [20]. The initial points were randomly selected to run the PMF model successively. By calculating the ratio of Q_robust_ to Q_true_ for different numbers of factors [44], the most stable five-factor source scheme was identified (Figure 5).

Factor 1 was weighted primarily on As (71.1%). The sources of the As in the soil included natural and anthropogenic sources [45]. The As contents in the study area were all within the national risk screening value, and the coefficient of variation was less than 50%, indicating that the possibility of human influence was weak. In general, some arsenic-bearing sulfide and oxide rocks migrate into soil via weathering and rain erosion [46], and the As contents of natural soil range from 0.2 mg/kg to 40 mg/kg. Therefore, Factor 1 can be concluded to represent the parent material, i.e., a geological source [47].

Se stood out, with an 84.9% contribution from Factor 2, followed by the contributions of Pb (38.9%) and Zn (35.8%). The Se concentration of seawater is high, and atmospheric dry and wet deposition are the primary mechanisms by which Se enters soil [48]. The study area is located on the east coast of the China Sea, with a 21 km-long coastline and relatively high Se contents. The decrease in the topography from west to east is conducive to Se deposition and enrichment. Additionally, the Pb in vehicle exhausts accumulates in the soil through atmospheric deposition [49], and the abnormal Zn contents may have been caused by dust produced by tire wear on the roads [50]. Based on the above analysis, Factor 2 can be concluded to be the atmospheric deposition source.

Factor 3 consisted predominantly of Cd (98.2%). Given that the coefficient of variation was greater than 100%, the Cd principally originated from anthropogenic activities. A large number of studies have shown that soil Cd pollution can be derived from agricultural activities and transportation emissions. A large amount of Cd exists in phosphate-compound fertilizers [51,52]; however, considering the low application of phosphate compound fertilizers in the study area, agricultural sources were excluded. Alternatively, car body wear, braking, and leakage of lubricating oil [53] lead to Cd diffusion and settlement in soil, and eventually form zonal soil pollution on both sides of the road [54]. The dense road network led to considerable Cd pollution in the study area, so Factor 3 can be concluded to be a traffic emissions source.

Factor 4 contributed significantly to Hg pollution (78.4%). Further analysis revealed that the Hg contents of the agricultural land and construction land in the study area were roughly the same. In agricultural activities, the irrational application of fertilizer containing Hg will lead to the direct entry of Hg into the soil, resulting in non-point source pollution [55]. In addition, fossil fuel combustion and metal smelting in industry are the most common sources of Hg [56,57]. There were certain numbers of coal enterprises and metal smelting enterprises in the study area, and heavy metals from combustion will inevitably enter the soil. In summary, Factor 4 can be concluded to be a mixed source of agricultural activities and industrial emissions.

Factor 5 contributed 79.39%, 88.59%, 58.52%, 90.14%, 90.59%, and 78.84% of the Cr, Ni, Cu, Co, V, and Mn pollution, respectively. Among them, the variabilities of the Co, V, and Mn contents were weak, so they were primarily affected by natural factors. The coefficient of variation of Cu was moderately strong, indicating both natural and anthropogenic effects, but the effect of industry was more prominent [51,58]. In addition, Ni and Cr were obviously enriched in the study area and exhibited extremely strong variability. Their contents were much higher in construction land than in agricultural land. The Ni content of petroleum is 1.4–64 ppm, so Ni pollution mainly originated from fossil fuel combustion. Cr primarily originated from wastewater and the dust discharged by industrial enterprises, such as smelting factories, electroplating factories, and machinery factories [59,60]. There were dozens of machinery manufacturing, petrochemical, and metal smelting enterprises distributed around the sampling sites. Therefore, Factor 5 can be concluded to be a mixture of geological and industrial sources.

### 3.4. Source Apportionment Based on the Unmix Model

The original data were processed and 221 soil sampling points were obtained after removing the outliers. After implementing the Unmix model, the distribution of the normalized values was plotted as red lines ranging from the normalized minimum to the normalized maximum values in Figure 6. The black lines in Figure 6 denote the normalized median values. Figure 6 indicates that the distribution boundaries of the 10 heavy metals were good, so they could be used in the analytical source model. Four major sources of soil heavy metals were analyzed using the Unmix model. The MinRsq of the model was 0.86, representing 86% of the variance of the species that could be explained, which was greater than the minimum value required by the system (MinRsq > 0.8). The MinSig/Noise was 2.27, which was higher than the minimum value required by the system (MinSig/Noise > 2), so the analytical results obtained using these four sources were reliable. The average concentration of each heavy metal in the four sources is shown in Figure 7.

The V content of Source 1 was the highest. According to the results of the descriptive statistics and pollution assessment, the coefficient of variation of V was low and so was the pollution level, indicating that V was mainly from natural processes. In addition, the Hg, Ni, and Cr contents of Source 1 were also relatively high. Based on the above analysis, it can be concluded that the Hg pollution was generated by agricultural and industrial activities, while the Ni and Cr stemmed from industrial pollution. Thus, Source 1 can be concluded to be a mixture of geological, agricultural, and industrial sources.

The contribution of Source 2 to the Zn pollution was significantly higher. Zn is an important additive in the production of automobile tires, and it is used for tire stiffening [21], lubrication improvement, oxidation resistance, and cleaning [61]. Moreover, Zn and its compounds are also contained in the particulate matter emitted by vehicles, and Zn-containing rubber fragments and dust can cause contamination when they enter the soil [62]. Transportation emissions also contain Pb, and the Pb concentration of Source 2 was high [63,64]. It should be noted that, although China banned the use of leaded gasoline in 2000, this heavy metal may still remain in the soil because it is characterized by difficult migration [52]. Therefore, Source 2 can be concluded to be a transportation emissions source.

In the source component spectrum of Source 3, the As and Mn contents were much higher. According to the descriptive statistics results, the degrees of As and Mn pollution were mainly no pollution, and their coefficients of variation were low. Source 3 is concluded to be a geological source.

The composition of Source 4 was similar to that of Source 1, i.e., high V, Ni, and Cr contents, indicating geological and industrial pollution sources. Unlike Source 1, the contributions of Source 4 to Mn and Co pollution were higher. Mn can be derived from industrial activities [65]. A large amount of Co is released during coal combustion [66]. Admittedly, the Mn and Co pollution generated by industrial activities was not serious in the study area. In general, Source 4 can be concluded to be a mixture of geological and industrial sources.

The contributions of Sources 1–4 identified using the Unmix model were 14.27%, 4.76%, 14.7%, and 66.28%, respectively. Source 1 and 4 together accounted for 80.55%, revealing that more than 80% of the heavy metal pollution in the study area may have been related to industrial activities, while traffic emissions had the smallest influence on heavy metal pollution. These results suggest that the discharge of industrial waste was still the main source of the soil heavy metal pollution in the study area, and it should also be the focus of pollution prevention and control measures.

The source apportionment results of the PMF and Unmix models were consistent overall (Table 3). The PMF model identified the sources of all 12 heavy metals and had a more potent explanatory power than the Unmix model. Moreover, the source types were more independent and precise. The Unmix model distinctly analyzed the sources of nine of the elements, with fewer pollution sources and a higher repetition rate. By incorporating the two models, it was concluded that As and V originated from natural sources; Se originated from atmospheric deposition; Cd, Zn, and Pb were derived from traffic emissions; and Cr and Ni were negatively affected by industrial pollution. The other four heavy metals came from multiple sources. Hg came from industrial and agricultural activities. Cu, Co, and Mn were mainly derived from geological and industrial sources, and Cu mainly came from industrial sources. The Co and Mn were mainly derived from natural sources.

## 4. Discussion

### 4.1. Effects of the Types of Industrial Enterprises on Soil Heavy Metal Pollution

Ningbo is a typical industrial city, with machinery, metal, and textile as the leading industries. Figure 8 shows the distributions of the enterprises above a designated size. The methods of soil heavy metal pollution source apportionment used in this study confirmed that industrial emissions were one of the inevitable sources in the Zhenhai District. In this section, we will discuss how the types of industry affected the soil heavy metal contents in Ningbo in depth.

In this study, radiation ranges of 500 m, 600 m, 700 m, and 800 m from industrial enterprises were used to establish buffer zones, and the average concentration of each heavy metal at the sample sites within the influence range was calculated. With the hope that the scope of influence would cover the study area as far as possible and that the sample repetition rate would be lower, 700 m was considered to be the most suitable scale for the study area. The results are presented in Table 4.

Except for the average concentration of Se around the industrial enterprises being roughly equivalent to the environmental background value, the average concentrations of the other 11 heavy metals exceeded their background values, indicating that the other 11 elements were all affected by industrial pollution to varying degrees. Indeed, a given type of industrial enterprise often caused the accumulation of multiple heavy metal pollution, exhibiting strong composite pollution characteristics [67].

The electrical appliances industry had the most significant accumulation effect for Cu. When untreated Cu enters water, it creates deposits that move downstream, and sewage irrigation causes the transfer of Cu into the soil and even into crops [68,69]. Chemical, metal, and coal enterprises can all be categorized as energy and raw material-processing industries, which not only require abundant raw materials and resource inputs, but also discharge large amounts of industrial wastewater, waste gas, and waste residue due to resource utilization efficiency and other reasons, so the pollution they generate is complex and diverse [70]. Among them, the influence of the chemical industry on the Zn, V, and Mn contents was far greater than those of the other industries. The enrichment of Cr and Ni by the metal industry was much higher, while the coal industry made large contributions to the Cd and Pb contents, especially Pb [65]. Furthermore, the paper industry had a significant influence on Hg and As accumulation in the Zhenhai District. There was a positive correlation between the wastewater produced by the paper industry and the As content of the soil samples [71]. As exists in the form of As anions in acidic soil and forms insoluble arsenide, so it has poor downward mobility and mostly exists in the surface soil [72]. White mud from paper making is an industrial solid waste [73], and it contains different forms of Hg, which may also enter the soil, resulting in Hg contents exceeding the standard. It should be noted that, although the machinery and textile industries did not directly contribute to the intensity of the pollution of specific heavy metals, these enterprises accounted for more than 50% of the enterprises in the Zhenhai District, and most of the heavy metal concentrations of the samples collected from the areas surrounding these enterprises exceeded the average concentrations in the study area. Therefore, their negative effects on the environment need to be considered and addressed.

### 4.2. Policy Implications including a Combination of Administrative Compulsion and Economic Incentives

The core purpose of source apportionment is to identify the pollution sources so that the pollution channels can be cut off from the sources and pollution can be effectively prevented and controlled.

Based on the results of the source apportionment, industrial discharge is the primary source of heavy metal pollution in Zhenhai District, Ningbo City. Therefore, to alleviate heavy metal pollution from industrial sources in the study area, it is of great significance to comprehensively control the total amount of heavy metal emissions from industrial enterprises. Ni, Cr, Zn, and Cd pose the highest risks in the study area, and their contents are primarily affected by the metal, chemical, coal, machinery, and textile industries. Therefore, to reduce heavy metal pollution in the short term, it is effective to comprehensively investigate enterprises of the relevant industries, and include enterprises that discharge large quantities of heavy metals in the list of key polluters according to laws and regulations. Their pollutant discharges should be strictly controlled, and production facilities that use outdated methods and cause serious pollution should be updated or shut down. It should be acknowledged that there is a contradiction between the development goals of the central government and the local governments in China [74]. The primary goal of local governments is to attain economic advantages, and they have an incentive to neglect the demand for environmental protection in exchange for local economic prosperity, which affects the effectiveness of policy implementation [75]. Consequently, it is necessary to adopt strict compulsory administrative measures.

However, the government must be aware that such coercive measures tend to increase the burden on financing cleaner production and waste treatment facilities for small and medium-sized enterprises, generating a contradiction regarding the unfair social distribution and increasing the cost of policy implementation [76]. Therefore, it is necessary to develop a new method of guiding industrial enterprises in order to proactively reduce the discharge of heavy metals. Zhang et al. pointed out that perfect and executable environmental policies need to satisfy two conditions: one is to internalize environmental externality, and the other is to minimize the cost of internalizing externality [77]. Specifically, to consummate the environmental tax collection system, it is suggested that the government of Zhejiang Province should include soil heavy metal pollutants in the collection scope and give full play to the reverse constraint and positive incentive mechanism of more discharge and more payment, less discharge and less payment, and no discharge and no payment. Furthermore, the government can select key enterprises in the above five industries to pilot cleaner production, demonstrating the economic and social benefits of the cleaner production of enterprises to other enterprises, which can contribute to the promotion of advanced cleaner production technology and application. This may result in this idea becoming the endogenous driving force of traditional industrial enterprise transformation and development [78].

Given that effluent is the inevitable route by which Cd, Cr, Ni, Zn, Pb, Cu, As, Mn, and V enter soil, the government can conduct dredging of river sections with high heavy metal concentrations in sediments to prevent toxic elements from entering the soil through irrigation [79].

The accumulation of Cd, Zn, and Pb in the study area was attributed to transport emissions. Governments can advocate the adoption of green travel methods, such as cycling and public transport, and promote the use of clean energy, such as wind, hydroelectric, tidal, and solar power [80].

Hg is another heavy metal with a relatively high degree of pollution in the study area. As one of its pollution sources, agricultural pollution also deserves the government’s attention. Relevant departments should strengthen the prevention and control of pollution from non-point agricultural sources, promote the reduction of the use of chemical fertilizers and pesticides, increase the use of new products, such as biological fertilizers, and curb the accumulation of Hg in the soil caused by sewage irrigation. Moreover, it is worth popularizing the improvement of the soil pollution management information database and long-term tracking surveys of the existing plots with heavy metal pollution. It is suggested that the basic information about these plots be obtained, and that their environmental risk and release over time be monitored and evaluated. Furthermore, limiting the exploitation of polluted plots and repairing them without the production of secondary pollution should be taken into consideration.

## 5. Conclusions

As a typical industrial city, the study area suffers from the problem of heavy metal pollution. Based on the assessment conducted in this study, nearly 70% of the soil in the study area was polluted by heavy metals. The 12 elements analyzed were affected by anthropogenic activities to varying degrees, and Zn, Cr, and Ni were the heavy metals that posed the greatest risks.

The different source apportionment methods employed produced roughly the same results. The PMF and Unmix models jointly identified geological, transportation, agricultural, and industrial sources, among which the geological and industrial sources played the most significant role. Based on the results obtained using these two models, Ni and Cr originated from industrial emissions; Cd, Zn, and Pb were from traffic emissions; As and V were from geological sources; Se was derived from atmospheric deposition; Hg synthetically stemmed from industry and agriculture; and Cu, Co, and Mn were from both geological and industrial sources; however, the Cu mainly originated from industry, while the Co and Mn were mainly from geological sources.

The results of source apportionment revealed that industrial emissions were the main source of soil heavy metal pollution in the study area and should be the focus of control and mitigation measures. The metal, chemical, coal, machinery, and textile industries had significant negative effects on the Ni, Cr, Zn, and Cd contents. In order to cut off the pollution channels from the sources, the government should implement compulsory administrative measures to control the total amount of heavy metal emissions from the above five key industries in the short term, and economic incentives, as well as administrative measures, should be implemented to provide enterprises with internal motivation to actively reduce emissions and achieve long-term sustainable development.

## Figures and Tables

**Figure 1 ijerph-19-03335-f001:**
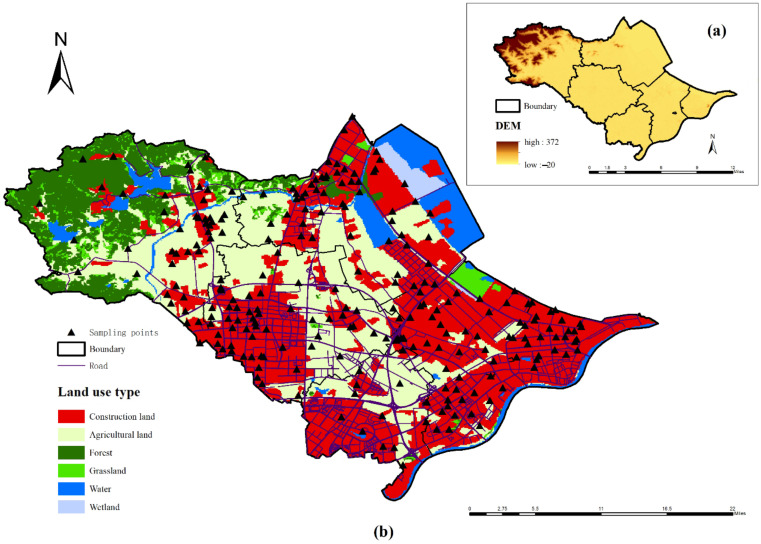
(**a**) Basic information illustrated by the digital elevation model and (**b**) distribution of the sampling points on the land-use type map of the study area.

**Figure 2 ijerph-19-03335-f002:**
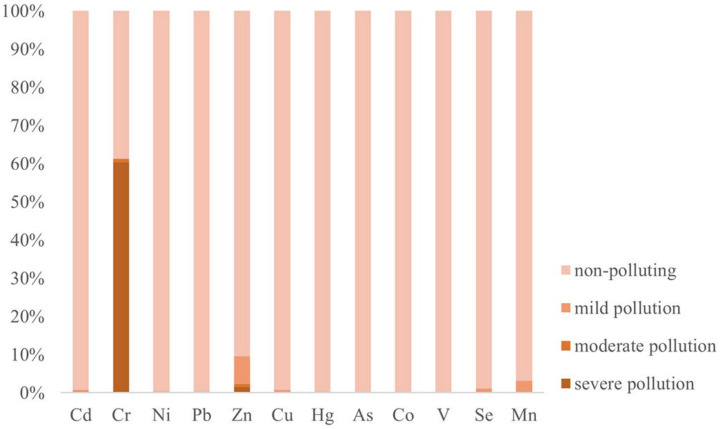
SPIs of the soil heavy metals in the study area.

**Figure 3 ijerph-19-03335-f003:**
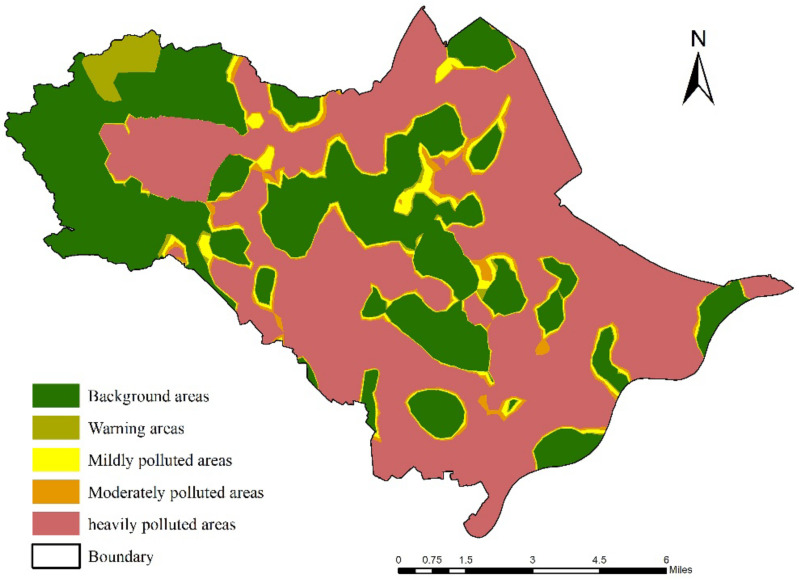
Comprehensive soil heavy metal pollution in the study area based on the NIPI.

**Figure 4 ijerph-19-03335-f004:**
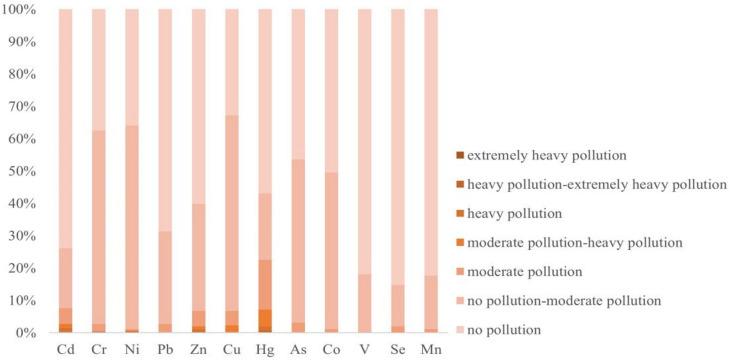
GI values of the soil heavy metals in the study area.

**Figure 5 ijerph-19-03335-f005:**
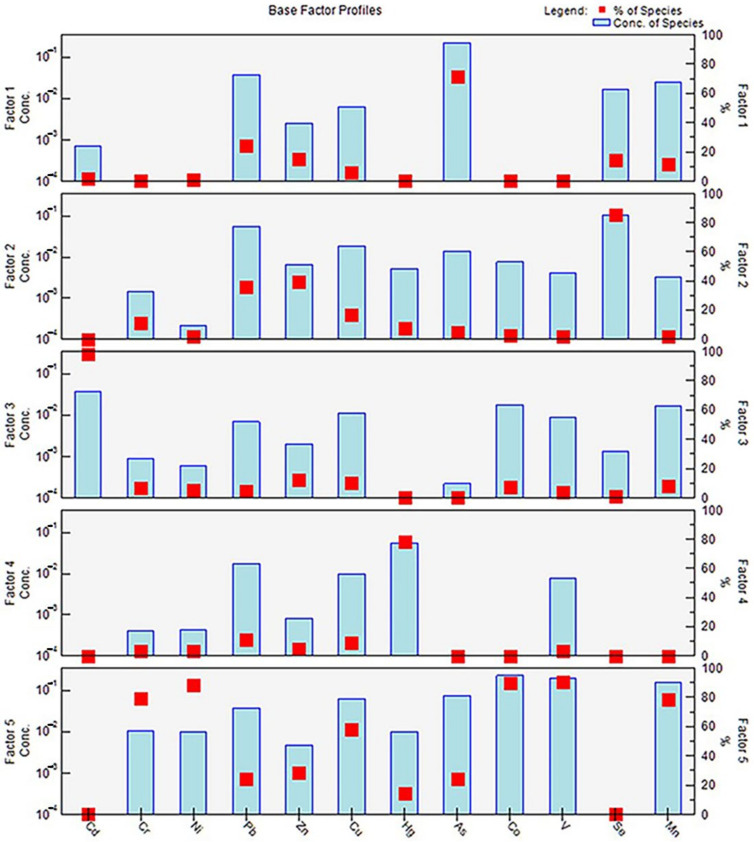
Source apportionment diagrams for the soil heavy metals based on the PMF model.

**Figure 6 ijerph-19-03335-f006:**
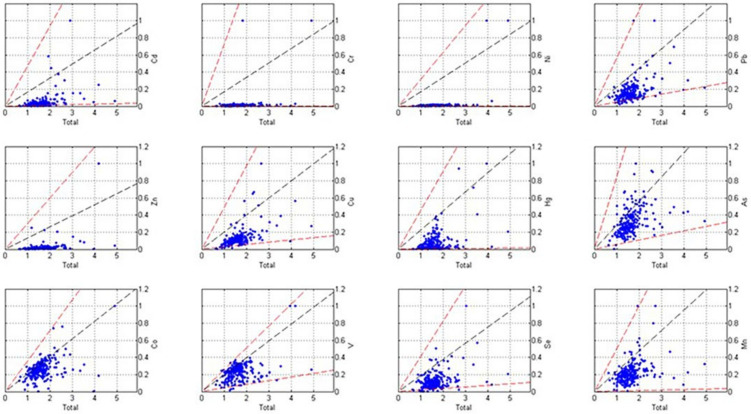
Geometric distribution diagrams of the heavy metal contents.

**Figure 7 ijerph-19-03335-f007:**
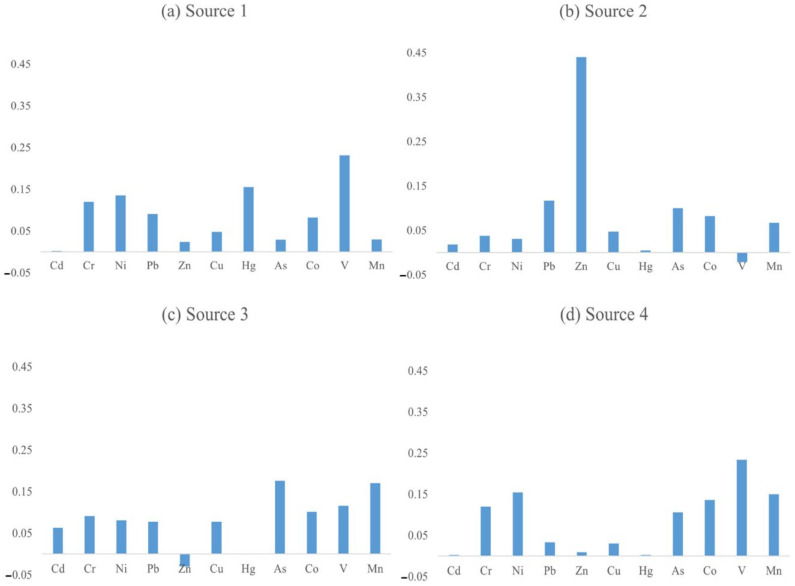
Source apportionment diagrams of the soil heavy metals based on the Unmix model.

**Figure 8 ijerph-19-03335-f008:**
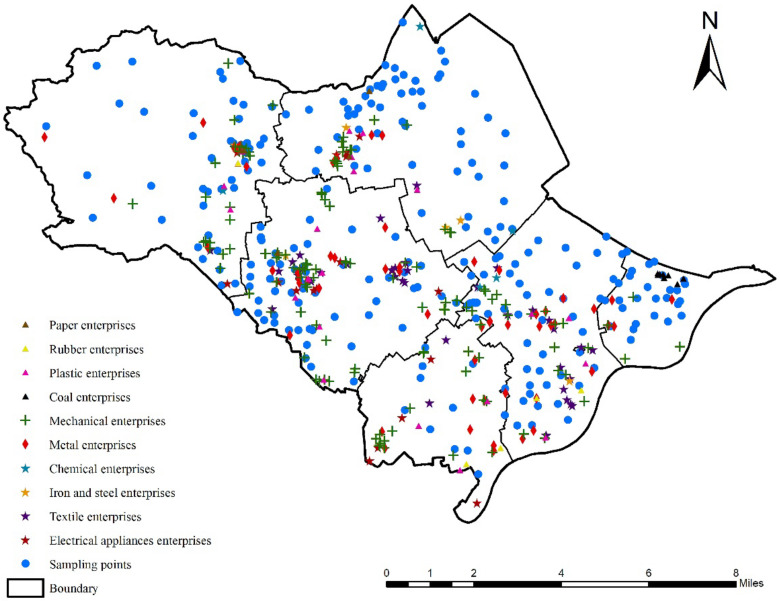
Map showing the distributions of the different types of industrial enterprises and the sampling sites.

**Table 2 ijerph-19-03335-t002:** Descriptive statistics of the 12 soil heavy metals in the study area.

Heavy Metals	Sample Numbers	Minimum (mg kg^−1^)	Maximum (mg kg^−1^)	Mean (mg kg^−1^)	Standard Deviation (mg kg^−1^)	Median (mg kg^−1^)	Skewness	Kurtosis	Coefficient of Variation (%)	National Risk Screening Values (mg kg^−1^)					Background Value (mg kg^−1^)
										Agricultural Land				Construction Land	
										pH ≤ 5.5	5.5 < pH ≤ 6.5	6.5 < pH ≤ 7.5	pH > 7.5		
Cd	248	0.014	4.52	0.2187	0.3892	0.124	7.333	70.21	178.00	0.3	0.4	0.6	0.8	65.0	0.14
Cr	248	12.6	4770	95.771	299.52	81.75	15.42	241.1	312.75	250.0	250.0	300.0	350.0	5.7	47.62
Ni	248	5.6	2580	45.45	162.22	36.4	15.47	242.2	356.92	60.0	70.0	100.0	190.0	900.0	21.51
Pb	248	18.4	187	46.187	17.85	42.25	3.411	21.75	38.65	80.0	100.0	140.0	240.0	800.0	31.62
Zn	248	55.4	4740	157.15	312.34	113.5	13.05	189.1	198.75	200.0	200.0	250.0	300.0	200.0	78.21
Cu	248	8.31	253	39.997	25.717	34.5	4.304	28.18	64.30	150.0	150.0	100.0	100.0	18,000.0	20.98
Hg	248	0.008	4.59	0.3511	0.5372	0.1655	4.762	32.82	152.99	1.3	1.8	2.4	1.0	38.0	0.15
As	248	2.03	24.4	9.0885	3.3277	8.405	1.399	6.345	36.61	40.0	40.0	30.0	20.0	60.0	5.4
Co	248	3.3	49.3	15.373	4.9647	15.2	1.903	13.73	32.29	70.0	70.0	70.0	70.0	70.0	10.15
V	248	29.6	376	107.48	32.088	111	2.049	21.29	29.85	165.0	165.0	165.0	165.0	752.0	87.21
Se	248	0.036	2.23	0.3062	0.2181	0.364	3.891	28.87	71.23	1	1	1	1	1	0.29
Mn	248	150	3070	771	338.85	721.5	2.293	13.7	43.95	1500	1500	1500	1500	1500	651.13

**Table 3 ijerph-19-03335-t003:** Comparisons of the source apportionments obtained using the PMF and Unmix models.

Method	Number	Pollution Sources	Main Heavy Metals
PMF model	1	geological source	As (84.9%)
	2	atmospheric deposition source	Se (84.9%), Pb (38.9%), Zn (35.8%)
	3	traffic emissions source	Cd (98.2%)
	4	agricultural and industrial sources	Hg (78.4%)
	5	geological and industrial sources	V (90.59%), Co (90.14%), Ni (88.59%), Cr (79.39%), Mn (78.84%), Cu (58.52%)
Unmix model	1	geological, agricultural, and industrial sources	V (58.33%), Hg (39.14%), Ni (34.09%), Cr (30.30%)
	2	traffic emissions source	Zn (333.33%), Pb (88.64%)
	3	geological source	As (43.14%), Mn (41.67%)
	4	geological and industrial sources	V (12.72%), Ni (8.37%), Mn (8.15%), Co (7.39%), Cr (6.52%)

**Table 4 ijerph-19-03335-t004:** Average concentrations of soil heavy metals around different types of industrial enterprises.

Enterprise Type	Sample Counts	Cd	Cr	Ni	Pb	Zn	Cu	Hg	As	Co	V	Se	Mn
BV	/	0.14	47.63	21.51	31.62	78.21	20.98	0.15	5.4	10.15	87.21	0.29	651.13
Electrical appliances	52	0.23	89.46	37.35	51.61	209.08	58.17	0.36	8.65	14.83	104.44	0.32	806.42
Textiles	110	0.15	82.75	36.38	49.6	196.91	45.37	0.33	8.23	15	106.09	0.31	805.77
Iron and steel	21	0.17	71.3	34.76	49.89	178.37	42.72	0.38	10.1	15.76	102.69	0.28	752.19
Chemicals	15	0.19	85.96	46.82	43.45	439.3	50.93	0.37	8.34	15.51	131.14	0.23	865.87
Machinery	366	0.18	85.76	36.99	51.3	170.74	50.07	0.4	8.91	15.43	107.93	0.32	773.68
Metals	153	0.23	114.34	51.76	50.92	186.31	53.39	0.37	9.05	15.22	102.31	0.32	815.94
Coal	35	0.24	70.5	28.28	56.68	115.49	43.89	0.18	9.88	14.32	87.93	0.37	851.8
Plastics	56	0.16	80	34.97	49.02	156.75	51.12	0.33	9.19	14.25	102.69	0.31	684.04
Rubber	16	0.22	77.01	33.39	53.87	137.71	46.28	0.33	8.14	14.85	109.14	0.26	669.75
Paper	29	0.2	70.88	31.74	50.53	223.93	46.73	0.41	10.66	14.67	93.77	0.35	722.55

BV: background value.

## Data Availability

No new data were created or analyzed in this study. Data sharing is not applicable to this article.

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
