# Peer review of "Pollution Assessment and Source Apportionment of Soil Heavy Metals in a Coastal Industrial City, Zhejiang, Southeastern China"

_ijerph, 2022, doi:10.3390/ijerph19063335_

Round 1

Reviewer 1 Report

Dear authors,

The manuscript presents a good study and discusses a very important and serious concern related to pollution of soils due to the heavy metals. The results seem to be interesting but the manuscript needs a major revision before it can be accepted for publication.

You are requested to spend more time on this article to make it useful for the readers. The study could be a guide for the policy makers to reduce the hazard of pollution in the study area but for that the quality of language must be improved.

The study is overall good but the results need to be properly presented and discussed. You should be very careful with the terminology. Many terms mentioned in the text or ambiguous. 

There are many grammatical errors and ambiguous statements. 

The study is of heavy metals. But no where in the text, the names have been mentioned. Symbols of the metals have been used since beginning. 

I have given my comments in the attachment.

Author Response

Dear Review,

Thanks for your advise. We have given our response in the attachment.

Reviewer 2 Report

The framework was well designed and the objectives of the work were well described. The results were well expressed and the discussion was relatively good. This research is worth publishing and can be accepted after minor corrections.

  • In the discussion, focus only on your own results.
  • Check the validity of the references
  • Although the English are satisfactory in general, a revision of the English is mandatory
  • The results of the study have not been discussed properly. Discussion part must be strengthened with the support of quality citations

Author Response

(The authors gave the same response as above.)

Reviewer 3 Report

GENERAL IMPRESSION

This paper clearly expresses and discusses characteristics of soil contamination in the study area while dealing with a lot of soil samples and locations of enterprises. Although scientific novelty is not high, pros and cons of PMF and Unmix models were also assessed as both models were applied in the study. Please refer to the following comments:

GENERAL COMMENTS

  1. Please consider again and discuss whether it is appropriate to apply mixed sources as a factor in PMF and Unmix models. For example, in the results of PMF (Lines 281–283), please consider again between ‘mixed source of agricultural activities and industrial emissions’ and ‘single source of agricultural activities or industrial emissions.’ I suggest this because each factor (pollution source) is ideally an independent one.
  2. Structure and flow of sentence as well as grammar should be extensively revised throughout the manuscript by a native. Mistypes should also be checked and revised (e.g., Lines 94, 207, 323). Use of proper noun should also be considered (e.g., Line 310 vs 318). Spacing should also be checked (e.g., Line 367).

SPECIFIC COMMENTS

  1. Line 55: The complexity comes from not only ecological processes, but also geochemical processes.
  2. Line 268: ‘a large amount of Cd’ does not seem to be adequate because the agricultural sources were excluded.
  3. 7: I recommend that the same range of values is applied in the y-axis for comparison.
  4. Line 480: As heavy metals also include Ni, Cr, Zn, and Cd, it should be changed to another expression.

Author Response

(The authors gave the same response as above.)

Round 2

Reviewer 1 Report

Dear authors

I have gone through the revised manuscript. It seems you have incorporated my suggestions and comments.

However, the language can still be improved.

Author Response

Dear Reviewer,

Thanks for your advise. We have given our response in the attachment.
